# Age-Related Differences in Stepping Reactions to a Balance Perturbation: A Functional Near-Infrared Spectroscopy and Surface Electromyography Study

**DOI:** 10.3390/brainsci12111479

**Published:** 2022-10-31

**Authors:** Ren Zhuang, Shizhe Zhu, Youxin Sui, Mengye Zhou, Ting Yang, Chaolan Wang, Tianjiao Zhang, Jin Wang, Chaojie Kan, Ying Shen, Tong Wang, Chuan Guo

**Affiliations:** 1Department of Rehabilitation Medicine, Changzhou Dean Hospital, Changzhou 213002, China; 2Department of Rehabilitation Medicine, The First Affiliated Hospital of Nanjing Medical University, Nanjing 210029, China

**Keywords:** aging, balance, perturbation, fNIRS, sEMG

## Abstract

We sought to investigate age-related differences in stepping reactions to a sudden balance perturbation, focusing on muscle activity and cortical activation. A total of 18 older healthy adults (older group, OG) and 16 young healthy adults (young group, YG) were recruited into this study. A cable-pull instrument was used to induce a forward perturbation at the waist level among participants, who were required to take the right step to maintain their postural balance. The seven right lower-limb muscle activities during periods of compensatory postural adjustments (CPAs) were recorded by surface electromyography. At the same time, the signals of channels located in the prefrontal, temporal and parietal lobes were recorded by functional near-infrared spectroscopy (fNIRS) during the whole process. Integral electromyograms of the right peroneus muscle, gluteus medius, and lateral gastrocnemius muscles showed greater activity for the OG in the CPA periods. Two channels belonging to the right pre-frontal (PFC) and pre-motor cortex (PMC) revealed lower activation in the OG compared with the YG. These findings can help us to better understand the differences at the peripheral and central levels and may provide some suggestions for future neuromodulation techniques and other clinical treatments.

## 1. Introduction

Falls are a leading cause of injury and death among older adults and a significant public health issue. Approximately 30% of older community-dwelling people [1] and 50% of adults over 80 years of age [2] fall each year. The falls in older adults not only cause a loss of income for the patient and caregiver but also intangible losses of mobility, confidence, and functional independence. The aging process results in a number of musculoskeletal (e.g., deficits in strength/power performance, atrophy of type-II muscle fibers, in particular, osteoporosis) and neural (e.g., loss of sensory/motor neurons) deteriorations [3]. All of these could lead to a high risk of falls among the elderly. The mechanism behind the high risk of falls needs further investigation. In this study, we focus mainly on muscle and cortical activity.

An increasing number of studies have paid attention to the constant balance tasks, such as standing on one foot, or gait tasks, and found an impairment of performance in older adults [4]. The ability to respond to sudden perturbations is also critical to balance control. Strategies for balance control can be divided into “fixed-support” and “change-in-support” methods based on the strength of the perturbation [5]. Of particular importance in preventing falls are change-in-support reactions, which involve rapid stepping and grasping movements [6]. Previous studies have observed impairments of parameters such as the greater center of mass, less step length, and so on in the response to a sudden perturbation in older adults, which could lead to a high risk of falls [7,8].

Anticipatory postural adjustments (APAs) and compensatory postural adjustments (CPAs) are two primary types of postural adjustment performed by the central nervous system. Prior to expected body perturbations, the central nervous system uses anticipatory activation of muscles to preserve body equilibrium, which is defined collectively as APAs. After the perturbation, CPAs produce the activation of postural muscles and deploy strategies to help restore body balance [9]. Surface electromyography (sEMG) can be used to measure APAs and CPAs. Previous studies have investigated age-related differences in APAs and CPAs under the known condition. At the same time, there are still some studies investigating their differences in response to an unpredictable perturbation. From these studies, the unknown occurrence or magnitude of a perturbation could lead to no or unadjusted muscle activity during the period of APAs. Claudino et al. [10] compared APAs and CPAs under both known and unknown conditions between young and old adults. Researchers utilized a pair of glasses with covered its lens to make participants blind to the occurrence of a hit of an accelerometer. They only found muscle activity during the period of CPAs was higher in older adults than young adults under the unknown condition, while there was no difference and even rare signals emerged during the period of APAs. In a study by Quinzi et al. [11], anticipatory perturbation was self-paced and externally triggered. From its line graph, no muscle activity was produced before the occurrence of the perturbation and there was still a delay after the perturbation. However, anticipated muscle activity was found under the self-paced condition. Xie et al. [12] made the magnitude of the perturbation unknown and found no different muscle activity during APAs under unknown conditions. Santos et al. [13] also proved this phenomenon. Therefore, CPAs, not APAs, play a key role when facing an unpredictable perturbation. Perturbations in daily life seem to be unpredictable. Based on a previous study [6], we modified some parameters of unknown conditions which were closer to daily life to explore the different patterns of muscle activity in the lower extremity in this experiment.

During the whole stepping process, proper engagement of the cortical cortex is also crucial to the balance equilibrium. The regions of the prefrontal cortex (PFC), pre-motor cortex (PMC), and primary motor cortex (M1) are all believed to participate in the regulation of balance control [14]. The reduced connectivity between cortical regions in older adults is accompanied by a worse performance of balance scales [15]. A previous study reported the involvement of the PFC in healthy participants when facing translation of the platform [16]. However, no additional studies have investigated the age difference in cerebral cortex involvement after a sudden perturbation. 

Among the various neuroimaging techniques that exist, functional near-infrared spectroscopy (fNIRS) may be the most suitable tool to measure such trials with greater locomotion. fNIRS is a non-invasive optical functional imaging system with ≥2 different frequencies between 700 and 900 nm that are sensitive to the detection of oxyhemoglobin (Oxy-Hb) and deoxyhemoglobin (Deoxy-Hb) [17]. The absorption coefficients of Oxy-Hb and Deoxy-Hb are different and their amounts can be evaluated using the modified Beer-Lambert law [18]. The phenomenon of increased activation of the cortex accompanied by the spontaneous rise in blood flow in the cortical region is known as neurovascular coupling [19]. Therefore, these two parameters can be used to determine the level of activation of the cerebral cortex by fNIRS. Compared with functional magnetic resonance imaging (fMRI) and electroencephalography (EEG), fNIRS is less sensitive to the motion artifact and is applied in many aspects of more complex motor behaviors, which are more naturalistic tasks [20]. Therefore, we used fNIRS to record changes in the cortical cortex in this study.

Broadly, the aim of this study was to investigate age-related differences in the CPA parameters of the lower-extremity muscles and the cortical activation that arises when facing a balance perturbation. We hypothesized that older adults could exhibit higher CPAs compared with young people at the peripheral level but decreased cerebral cortex activation at the central level.

## 2. Materials and Methods

### 2.1. Participants

We recruited 20 young participants (12 men; mean age, 27.31 ± 4.94 years) into the young group (YG) and 20 older participants (8 men; mean age, 68.57 ± 5.48 years) into the older group (OG). The inclusion criteria were:(1)20 participants aged 20–40 years old, 20 participants over 60 years old;(2)Being able to stand independently for more than 10 min;(3)Right-handedness and leg dominance;(4)Family members and patients signing the informed consent.

The exclusion criteria were:(1)A history of neurological lesions, brain tissue damage, vestibular, visual or proprioceptive disorders;(2)Being suffering from malignant progressive hypertension, severe visceral system disease, and malignant tumor;(3)A history of organic brain disease, mental disorder, or epilepsy;(4)Cognitive and communication impairments;(5)Administration of drugs that affect balance function for 1 week before the test;(6)Complicated diseases that seriously affect the sensation and movement of the lower extremities, such as rheumatoid arthritis, lumbar disc herniation, lower extremity trauma, severe skin damage, fractures, and peripheral neuropathy.

Handedness, dominant leg, body mass index, Mini-mental State Examination score, and Berg Balance Scale score were recorded. Handedness was evaluated according to the Edinburgh Handedness Inventory. Leg dominance was determined by asking the participants “If you had to shoot a ball at a target, which leg would you use to kick the ball?” Only the dextral-domain participants were finally included in this study. Written informed consent was obtained prior to study participation from all participants. Four young adults and two older adults could not finish the experiment due to their fNIRS signals being too weak. The experimental procedure was approved by the Human Ethics Committee of Changzhou Dean Hospital, Changzhou, Jiangsu, China (CZDALL-2022-001) and was registered in ClinicalTrials.gov (ChiCTR2200059252). Demographic data and clinical characteristics are shown in Table 1.

### 2.2. Experimental Design

To address our research questions, we used a cross-sectional design wherein each participant’s testing took place during a single measurement session. We recorded sEMG and fNIRS signals simultaneously during posture control. The schematic diagram of this trial is shown in Figure 1.

#### 2.2.1. Postural Perturbations

According to a previous study [6], we adopted waist-pull perturbations to deliver the postural perturbations, aiming to evoke stepping (Figure 1). The anterior waist-pull was delivered by dropping a weight (20% of the participant’s body weight) attached to a belt worn around the waist via a cable and pulley system. The choice of 20% of body weight was based on a previous study [6] and we checked that this weight could induce the spontaneous step without extra actions. The pulley was fixed on a desk and could be adjusted in height to fit the participants. We set a curtain in front of the participants so that the perturbation could be triggered in an unpredictable way. On the other side of the curtain, the weight system was controlled by an experienced researcher. Before the release of the weight, the cable was straight, without tension, and the weight was released from a height of 40 cm over the ground to create the perturbation without warning the participant. We also arranged for another researcher to stand beside each participant to ensure their safety.

#### 2.2.2. Surface Electromyography

sEMGs were evaluated using a desktop DTS system (sample rate, 3000 Hz; Noraxon, Scottsdale, AZ, USA). sEMG was used to record the electrical activity of the following seven right lower-limb muscles: lateral gastrocnemius (LG), tibialis anterior, biceps femoris (BF), medial gastrocnemius, peroneus muscle (PM), gluteus medius (GM), and rectus femoris [21]. The skin was cleaned using a 75% surgical spirit before attaching the disposable EMG electrodes (Noraxon, Scottsdale, AZ, USA) to the belly. Then, wireless sensors were linked to the electrodes and affixed with medical tape. To define the moment of postural perturbation, we used a camera (Logitech C270, Logitech, Lausanne, Switzerland) to record the whole process and an accelerometer fixed on the belt to catch the instance of the perturbation.

#### 2.2.3. fNIRS Measurement

The fNIRS system (NirScan Danyang Huichuang Medical Equipment Co., Ltd., Zhenjiang, China) with wavelengths of 730 and 850 nm was used to detect cerebral hemodynamic changes during postural perturbation. Data were sampled with a frequency of 11 Hz. A total of 42 channels set up by 19 source optodes and 16 detector optodes were symmetrically positioned over the regions of the prefrontal, temporal and parietal lobes. The spatial locations of sources, detectors, and anchor points (located at Nz, Cz, Al, Ar, Iz referring to the standard international 10–20 system of electrode placement) were measured by an electromagnetic 3D digitizer device (Patriot, Polhemus, Colchester, VT, USA) on a brain template [22]. The acquired coordinates were then transformed into MNI coordinates and further projected to the MNI standard brain template using a spatial registration approach in NirSpace (Huichuang Medical Equipment Co., Ltd., Danyang, China). According to the MNI coordinates, we can obtain the location of each channel which shows how many percentages of different Brodmann’s areas. Then, we divided channels into regions of interest based on the highest percentage of Brodmann’s areas [23]. The distance between the detector and source was 30 mm to ensure propagation to the gray matter beneath the optodes. Before the optodes were fixed, hair was brushed away within every hole of the cap to ensure good light coupling. The details of channel attrition and division are shown in Table 2 and Figure 2.

#### 2.2.4. Experiment Procedures

After considering several pre-experiments and a summary of the fNIRS experimental research [24,25], we decided to use an event-related design to record the movement-induced changes in blood oxygen. In the beginning, a 30 s rest period was first allowed to collect the baseline data of each participant and all participants maintained an upright stance with their feet shoulder-width apart. Then, the researcher released the weight according to a pre-programmed signal (Figure 3) showing on the monitor, and participants maintained their balance by stepping on their right leg. Then, they kept the stepped-out posture for 25 s. After 25 s, the participants received the instruction ‘step back’ to return to the original upright stance and wait for the next perturbation for 30 s. There were five perturbations in total. Before the experiment, every participant was given 5 min to learn and adapt to the experiment and to make sure that they were familiar with the whole procedure. During the pretest, participants were just told what movements they should take (e.g., taking a right step, maintaining an upright stance, stepping back). However, the timing and the number of trials were unknown to all participants. All participants were unaware of the program prior to the test and each perturbation was a random event to them. Meanwhile, we made sure that 20% of their weight could lead to a stepping response, and then we told them to step on their right legs to maintain their balance before informal trials. The event design diagram is shown in Figure 3.

### 2.3. Data Processing

#### 2.3.1. sEMG Data Processing

An analysis software (MyoResearch 3.6; Noraxon, Scottsdale, AZ, USA) was used to filter (50–500 Hz), rectificate (all negative amplitudes are converted to positive amplitudes), and smooth (root mean square at 50 ms) the raw data of muscle signals. The time point at which the first apparent change in amplitude of the accelerometer signal occurred was regarded as the time point when the perturbation was generated, and it was counted as 0 ms (T0). According to a previous study [13], the period of CPAs was 50–300 ms. Then, we used data from these two periods to calculate the value of the integral electromyogram (iEMG) compared with the baseline (−1000 to −950 ms). The division method of this time window was supported by previous studies, and the related results showed that the period can effectively reflect the muscle activity of CPAs [26]. Results from five trials were averaged into one. The equation of CPAs is shown in Equation (1):(1)$$\int EMG=\int_{+300}^{+50}EMG-5\int_{-1000}^{-950}EMG$$

#### 2.3.2. fNIRS Data Processing

We used the NirSpark software package (HuiChuang, Zhenjiang, China) to analyze fNIRS data. Data were pre-processed using a series of steps, as follows. First, the raw NIRS light intensity was converted to an optical density (OD) signal. Second, a cubic spline interpolation method was used to detect and correct motion artifacts caused by head movement during data acquisition (parameters were set as the time range where to check the indicative of a motion artifact (tMotion) = 1 s; the threshold for ratio of standard deviation of input signal over time range tMotion (STDEVthresh) = 15.0; and the threshold of input signal over time range tMotion (AMPthresh) = 0.5). Third, a bandpass filter with cutoff frequencies of 0.01–0.1 Hz was used to remove physiological noise (respiration, cardiac activity, and low-frequency signal drift) [27]. Finally, the filtered OD signal was converted to the blood oxygen concentration by applying the modified Beer-Lambert law using a path length factor of 6 [28]. The extinction coefficient of HbO is 1174.53 and 2526.39 by the infrared wavelength of 730 and 850 nm and of HbR is 2985.83 and 1798.64, respectively [29].

After data pre-processing, we used Oxy-Hb as our primary indicator in the activation analysis because the Oxy-Hb signal generally has a better signal-to-noise ratio than that of Deoxy-Hb [30]. The mark of the perturbation was pre-programmed and defined as 0 s. The duration of instant stimulation was set as 1 s. We set “0–25 s” as a block duration for the hemodynamic response function and “−2 to 0 s” as the reserved baseline state. The block average was generated by superimposing and averaging the blood oxygen concentrations collected from each of the five paradigms. For each pre-treated experimental dataset, a generalized linear model was used to analyze the HbO_2_ time series data. Each channel’s *β* value of all participants in the two groups was obtained. The *β* value, which reflects the level of cortical activation of the channel, was used as an estimate of predicting hemodynamic response functions of the HbO signal to represent the peak value of the hemodynamic response function [31].

### 2.4. Statistical Analysis

SPSS version 23.0 (IBM Corporation, Armonk, NY, USA) was used for the statistical analysis of the sEMG and population characteristics. First, the Shapiro–Wilk test was performed to determine the conformity to the normal distribution. If the data passed the Shapiro–Wilk test, a two-sample *t*-test was used to detect significance; otherwise, the Mann–Whitney U test was used for analysis. The data of fNIRS were also assessed on the NirSpark software package (HuiChuang, Zhenjiang, China). A two-sample *t*-test was performed on the fNIRS data, and *p* values were adjusted by false-discovery rate correction. All significance levels were set at 0.05.

## 3. Results

Demographic and scores of relevant scales for the OG and YG are presented in Table 1.

### 3.1. iEMG of CPAs

All details of CPAs are shown in Table 3. There were significant differences in the CPAs of the LG (*p* = 0.001, *t* = 3.662), PM (*p* = 0.003, *t* = 3.225) and GM (*p* = 0.020, *U* = 77), and higher iEMG amplitudes were similarly recorded in the OG. The remaining muscles showed no significant differences in CPAs between groups.

### 3.2. β Value of Cortical Activation

All details of *β* values are shown in Figure 4 and Appendix A. Only channel 30 located in the right PFC (adjust-*p* = 0.033, *t* = −3.489) and channel 31 located in the right PMC (adjust-*p* = 0.033, *t* = −3.743) had significant differences, and the activations of these two channels in the OG were lower than those in the YG. The remaining channels showed no significant differences.

## 4. Discussion

To our knowledge, few studies explored the age-related differences in the cerebral cortex after a balance perturbation. The results of sEMG showed that the amplitudes of CPAs among the elderly were greater than those of the young in some key muscles. Another finding was that activation of channels in the right PFC and PMC was impaired in older adults compared with younger ones. This also suggests that age-related differences during balance recovery exist at both peripheral and central levels.

### 4.1. Differences in sEMG

Differences in CPAs existed between the OG and YG, which were supported by sEMG. We observed increased muscular activity in some of the dorsal and ankle muscles in the OG during the CPA periods.

The attribution of the muscles that showed significant differences met our expectations, and the direction of the forward pull was responsible for this. For the biomechanical considerations, the leg muscle contraction is resisted by an equal-sized reactive force from the surface, which can in turn be transmitted through the contracting dorsal muscles [32]. In a previous study [33], the contraction of the dorsal muscles was also proved to resist anterior perturbation for maintaining postural balance. The purpose of APAs and CPAs is to minimize the adverse effects of the disturbance and restore postural balance [9]. Therefore, increased activation of the LG, GM and PM for OG could also be shown in the periods of CPAs.

Regarding increased muscular activity, despite differences in experimental design, previous studies have reached similar conclusions in CPAs. Claudino et al. [10] found that, compared with the YG, the CPAs of some lateral, ventral, and dorsal muscles of the OGs (with and without falls) were significantly more robust when the right shoulder was influenced by weight, consistent with the findings of Lee et al. [34] and Kanekar et al. [35]. We believe there are several reasons for the increased CPAs. One reason may be the strength of the perturbation, which is mentioned above. In this study, we chose 20% of the participant’s weight. For older adults, the loss of skeletal muscle strength, mass, and quality is unavoidable [36]. A previous study also indicated that age effects may be considered a less efficient neuromuscular modular control strategy, manifested as higher muscle co-activity within each muscle synergy [37]. Therefore, this set of perturbations for the OG could be relatively stronger than that for the YG. Furthermore, Xie and Wang [12] compared sandbags of different weights to create a perturbation and observed the effect of this factor on the APAs and CPAs. We could see the tendency for CPAs to experience a synchronous rise with the rise of weights. Our participants were fully familiar with the intensity of the perturbation before the trials, so the OG may have more significantly increased their muscle activity compared with the YG to confront the perturbation. Another reason for the increased CPAs may be related to the employment of the “redundancy response strategy”, i.e., the maximum assumption of perturbation magnitude [12]. Central nerves cannot provoke an optimal response of the postural muscles, because it lacks necessary information on the magnitude or time of perturbation [10]. Therefore, the maximum assumption of perturbation would be anticipated in the unknown condition. We asked about the participants’ feelings after the experiment, and some of the OG expressed a fear of falling during the experiment, but there was little such feedback received from the YG. This phenomenon of older adults being more nervous when performing balance tasks was also found in another experiment [38]. The associated lack of confidence and high level of stress could contribute to the rise in muscle activity. Inefficient postural adjustments for OG when compared with YG (Lee et al., 2015) may be a reason for more CPAs in older adults. However, further research using more accurate measurements is needed in the future.

### 4.2. Differences in fNIRS

The instant reactions to the balance perturbation were thought to be largely regulated by subcortical networks, but more and more studies have proved that the cerebral cortex plays an important role when facing a postural challenge. Mihara et al. [16] were the first to apply neuroimaging tools to the online recording of cortical activation after a balance perturbation and provided direct proof of PFC’s involvement in human balance control. In a previous review [39], the cortical cortex was mentioned to modulate late-phase or change-in-support response characteristics perhaps through direct control. However, this area has not attracted widespread attention in the domain of neuroimaging, which is a classic paradigm used in the exploration of peripheral movement. As one reason for this trend, fMRI and EEG are sensitive to motion artifacts, and fNIRS can tolerate more artifacts in comparison [40]. This feature makes fNIRS more suitable for application in naturalistic situations. As another reason, the maintenance or repetition of some actions or tasks has attracted the greater observation of the change or activation of specific cerebral cortices, especially for the dual task, which is a popular method for investigating the increased risk of falls in the elderly. Studies have reported a similar conclusion that the activation of PFC in older people is enhanced more than that in young people when performing balance tasks [41,42,43]. This phenomenon is often explained by the theory of “compensation”, which is defined as the neural recruitment that enhances task performance in healthy aging [44]. In balance tasks, older people require more compensation from cognitive resources to maintain their postural balance [45]. However, this theory cannot explain our results, even contradicting our findings. In fact, the experimental design described above is a kind of block design, but our study is an event-related design and may have a different pattern for the activation of the cortex. The aim of a block study design in this context is to observe the resource allocation of the PFC in maintaining postural stability, while our experiment observed the regulation of change-in-support responses by the PFC after a perturbation.

The PFC is considered the main region regulating activity relevant to cognitive ability, such as attention, execution, and multitasking [46]. The attentional resource after a perturbation may have two main components: conflict detection during task performance and allocation of attention to adjust the attentional resources when necessary. It also accepts the peripheral sensory input and has a direct link with the PMC [47]. Therefore, it plays an important role in balance recovery during the whole process of the perturbation. In the study by Coelho et al. [48], fNIRS was also used to monitor step judgment in older and younger adults under different conditions. Changes in oxygenated hemoglobin in the PFC and supplementary motor area (SMA) were observed within 10 s after the event stimulation. It was found that changes in the SMA and PFC in the OG were both lower than those in the YG when there were incongruent options surrounding the right option. Although our study findings were not entirely consistent with those of this experimental design, this other study does demonstrate that the brain’s activation pattern in response to such stressor events is different from that of sustained tasks. The limit of PFC resources may also be seen in an emergency event, appearing as decreased activation. It is noticed that only the right PFC was decreased in our study. In Reuter-Lorenz’s study, they found only compensatory activation was existing in the right PFC when older adults perform at lower demand tasks. However, with increasing task difficulty, activated brain areas appeared underactivated at higher task demand. Compared with our task, the perturbation happened suddenly, so the OG could not use the compensatory activation to improve their performance. Meanwhile, the sudden perturbation may be more difficult for OG, so the phenomenon can be also seen in our study [49]. Furthermore, the influence of stress was also seen in the PFC. Tyagi et al. [50] used virtual reality to simulate an emergency environment to elicit stress, and participants were divided into a stress group and a control group. Ultimately, the stress group exhibited suppressed PFC activation during training. This phenomenon may be associated with reduced sensitivity of the PFC to external stimulations in the face of increased sympathetic activity and decreased parasympathetic activity under stress [51,52]. In addition, higher stress in OG could lead to a decreased activation on right PFC. The activation in the right ventrolateral PFC decreased in the more stressful condition as compared with the less stressful one in healthy subjects [53]. All of these indicate that stress may play a key role in the increased PFC.

In our study, we also observed decreased activation of the channel located in the right PMC. The PMC consists of the pre-motor area (PMA) and SMA. It is known that the SMA is involved in the planning and generation of internally driven actions, including APAs at gait initiation [54,55], and the PMA is involved in the sequencing of movements activated by external stimuli [56]. Except for the link with the PFC, the neural signals of the PMC were directly projected to the M1 and spinal cord to elicit an action [47]. These functions can be planned and executed proactively according to endogenous desires or in reaction to sensory stimuli. Externally generated movements are initiated by sensory input that is first conveyed to the frontal cortex and then prepared in more lateral motor regions of the PMC [57]. Frontal cortical regions, compared with other cortical areas, are disproportionately vulnerable to both age-related grey matter atrophy [58] and deterioration in white matter tract integrity [59]. This atrophy may be the reason for the reduced activation in both the right PFC and PMC after a balance perturbation. Less activation of cortices may indicate diminished regulation of motor performance. In accordance with our findings, Coelho et al. [44] also observed a decrease in SMA in the OG. It is worth noting that they also conducted a similar experiment on stroke patients, which revealed that the increase in HbO2 related to a postural disturbance in the unaffected SMA in stroke patients is significantly correlated with the increase in balance function as measured by the Berg balance scale.

Although we did not find a significant difference in SMC, the PFC, PMC, SMC, and subcortical networks are considered the connected circuit of balance control. The decreased activation of the right PFC and PMC may indicate reduced regulation of these cortices, leading to worse performance in balance control among older people.

### 4.3. Limitation

There are still some limitations in this experiment. The first is the division of regions. The spatial resolution of fNIRS is relatively lower than that of fMRI, and the Broadman area of the standard brain still cannot distinguish brain regions well. For example, PMC brain regions cannot be divided into PMA and SMA precisely. Therefore, we can only speculate roughly about the interpretation of channels in brain regions, and the evidence for the interpretation lacks certain confidence. The more precise and individualized coordinate of fNIRS was needed for the consistent and rigorous investigation. The second is the selection of peripheral and central indicators. Because CPAs are instantaneous indicators before and after the perturbation, the *β* value of fNIRS is the activation value calculated from the blood oxygen changes within a period after the perturbation. Therefore, the two correlations are irrelative on a time sequence, and future trials should consider how to choose more reasonable indicators to truly connect the central nervous system with the peripheral performance. Third, under the limitations of the device of sEMG, we had no more available sensors to record the muscle activity of bilateral lower limbs. Aside from the stepping leg, the supporting leg is also critical for the maintenance of posture and could have different patterns of muscle activity. Future studies need to further investigate this point. Meanwhile, they should consider more suitable musculoskeletal indicators to correlate muscle activity and brain activity. Otherwise, the Results of our study only showed reduced activation of the right PFC and PMC. Are there different functions of these brain areas between the right and left sides? Especially for the right PMC, few studies investigated differences between the two sides. Future studies are still needed to prove the differences. Finally, this study only enrolled healthy older adults, and therefore, BBS could not be a sensitive tool to detect balance disorders in older adults. The findings of our study may only exist in older adults with full BBS scores. Overall, further investigations are warranted to clarify the difference in stepping response to a sudden perturbation in older adults compared with young people.

## 5. Conclusions

Older adults have a high risk of falls and the instant response to the loss of equilibrium is very important to prevent the occurrence of falls. The mechanism behind the worse response is shown at both the peripheral and central levels. In addition to obtaining findings such as those of a prior study regarding CPAs, we also observed age-related differences in activation of the PFC and PMC, which declined according to advancing age. Such findings can help us to better understand the mechanism behind the worse response to a sudden perturbation. In the future, considering that the elderly would maintain their own balance by enhancing muscle activity, further strengthening of posture muscles may play a role in reducing the risk of falls. Moreover, the right PFC and PMC could be two key targets for central nervous modulations to help older adults improve their balance control.

## Figures and Tables

**Figure 1 brainsci-12-01479-f001:**
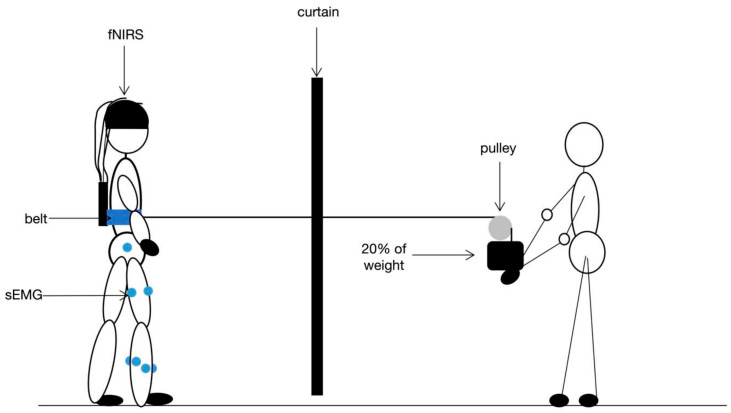
The schematic diagram of the experiment.

**Figure 2 brainsci-12-01479-f002:**
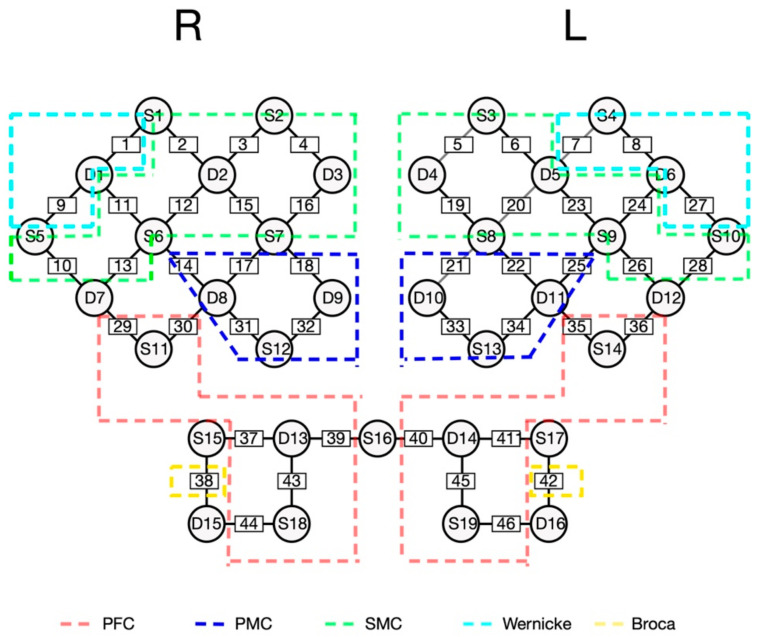
The channel attrition of fNIRS. PFC = pre-frontal cortex; PMC = pre-motor cortex; SMC = sensorimotor cortex.

**Figure 3 brainsci-12-01479-f003:**
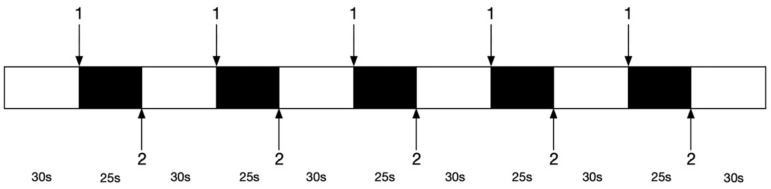
The event design diagram. Mark 1 means the signal of the perturbation and Mark 2 means the signal of returning.

**Figure 4 brainsci-12-01479-f004:**
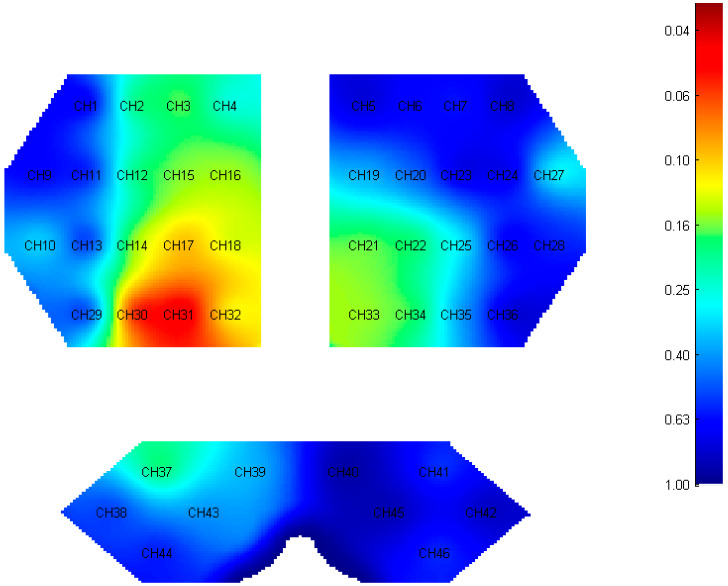
Activation of all channels. The value is represented by adjusted *p* for *β*. The smaller value is closer to red, otherwise, the color is closer to blue.

**Table 1 brainsci-12-01479-t001:** The characteristics of all participants. The first values represent group means while the values in the brackets stand for their standard deviations.

	OG	YG	*p*
Age (year)	68.566 (5.480)	27.313 (4.936)	<0.001
Sex (male/female)	8/10	12/4	NA
BMI (kg/m^2^)	23.639 (3.242)	23.873 (3.154)	0.833
MMSE (score)	28.222 (1.957)	30.000 (0)	0.001
BBS (score)	56 (0)	56 (0)	NA

OG = older group; YG = young group; NA = non-applicable; BMI = Body Mass Index; MMSE = Mini-mental State Examination; BBS = Berg Balance Scale.

**Table 2 brainsci-12-01479-t002:** The characteristics of all participants.

ROI	Channel
Right PFC	29, 30, 37, 39, 43, 44
Left PFC	35, 36, 40, 41, 45, 46
Right PMC	14, 17, 18, 31, 32
Left PMC	21, 22, 25, 33, 34
Right SMC	2, 3, 4, 10, 11, 12, 13, 15, 16
Left SMC	5, 6, 19, 20, 23, 24, 26, 28
Right Wernicke	1, 9
Left Wernicke	7, 8, 21
Right Broca	38
Left Broca	42

ROI = region of interest; PFC = pre-frontal cortex; PMC = pre-motor cortex; SMC = sensorimotor cortex.

**Table 3 brainsci-12-01479-t003:** The result of iEMG of CPAs (V). The first values represent group means while the values in the brackets stand for their standard deviations.

	OG	YG	*p*	*t*/*U*
LG	16.363 (6.754)	9.289 (4.380)	0.001	3.662
TA	16.748 (10.402)	12.954 (6.294)	0.211	107
BF	10.093 (5.832)	6.560 (4.570)	0.06	1.949
MG	18.807 (13.928)	15.804 (10.214)	0.551	126
PM	15.172 (8.013)	7.640 (5.080)	0.00	3.225
GM	4.253 (3.032)	2.788 (3.279)	0.020	77
RF	8.432 (6.055)	8.904 (6.793)	1	144

LG = lateral gastrocnemius; TA = tibialis anterior; BF = biceps femoris; MG = medial gastrocnemius; PM = peroneus muscle; GM = gluteus medius; RF = rectus femoris.

## Data Availability

Data can be made available by the corresponding author upon request.

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
