# Peer review of "Age-Related Differences in Stepping Reactions to a Balance Perturbation: A Functional Near-Infrared Spectroscopy and Surface Electromyography Study"

_brainsci, 2022, doi:10.3390/brainsci12111479_

Round 1
Reviewer 1 Report
The topic of this paper is timely. The paper is also well-structured and well-written. The authors should address the following comments to improve the paper.
1) In Figure 2: It is not clear how the brain areas were identified and optodes were placed on the subjects. The reference is taken (e.g. Cz, C3, C4, etc) to place the optodes that must be indicated in this figure. Also, the optodes were placed on the hit and trial method, or some equipment was used to identify the reference point on each subject. Which reference system was used to place the optodes? More explanation is needed for a clear understanding of the reader.
2) In Figure 3: The time scale should be added instead of mentioning a and b.
3) More detail is required for the conversion of OD to HbO and HbR. Include the used extinction coefficient used for the conversion. Why path length factor of 6 was chosen for both wavelengths? Kindly justify or add a reference.
4) The authors should proofread the manuscript. Lines 243 to 257 are journal guidelines and redundant. Remove these lines.
Author Response
Thank you for your all questions! Our responses are as followed.

Reviewer 2 Report
Thank you for the opportunity to review the present manuscript. The manuscript was well written and examined a novel area. Below you will find some suggestions, which may improve the quality of the manuscript.
1. Abstract
- Implications of the EMG and fNIRS findings would be well suited for abstract
2. Introduction
-The introduction appropriately describes APAs, CPAs, and why fNIRS is suitable for this study. I would encourage the authors to add information on age-related differences in the musculoskeletal and neurological systems, and how these differences may contribute to balance issues and falls.
3. Methods
-There is a discrepancy in the number of participants. Section 2.1 states 20 participants in each group, Table 1 states a different number. Please clarifity.
-Methods were appropriately described. I appreciated the use of explanation and clarity, particularly in Section 2.3.
-Not sure if lines 243 - 257 were meant to be included.
4. Results
-The results are clear and succinct.
5. Discussion
- The discussion needs more on age-related differences in the musculoskeletal and neurological systems, and how these differences may contribute to the findings
-The discussion needs more regarding future directions and potential clinical implications of the findings.
Author Response
Thank you for your all questions! Our responses are in the following file.

Reviewer 3 Report
This study investigated that difference of response in healthy young subjects and older subjects using the balance task using sEMG and fNIRS. As a results, healthy young subjects and older subjects were difference of response in sEMG and brain activity. The methodology for this study was not clear. In addition, there were many careless errors.
Materials and Methods
3 of 14 122-128. What did author determine that fNIRS signals too weak?
3 of 14 Table 1. Please change Berg to BBS.
3 of 14 What is *?
3 of 14 Table 1. What is unit? Please add the unit.
4 of 14 136-149 What instructions did participants receive?
4 of 14 136-149 The sEMG only measures the right foot. Why is that? Did all participants step on their right foot? Did any of the subjects step on their left foot? I thought the task of this experiment was not clear.
4 of 14 136-19 Author said ‘’ We set a curtain in front of the participants so that the 143 perturbation could be triggered in an unpredictable way.’’ How did you define the timing? Depending on the method of induction, the subject could have been predicted to some extent.
5 of 14 172 What did author base ROI on? Do you use NIRS-SPM? If not so, this ROI was not correct the brain area. Please refer to this prior research.
1. Kumai K, et al. Brain and muscle activation patterns during postural control affect static postural control. Gait & Posture, 2022, 96; 102-108.
2. Sakai K, et al. Comparison of Functional Connectivity during Visual-Motor Illusion, Observation, and Motor Execution. Journal of Motor Behavior. 2022, 54(3), 354-362.
7 of 14 243-257 What is this?
Result
8 of 14 Is there a correlation between muscle activity and brain activity?
Discussion
10 of 14 In this study, right PFC and PMC were decreased. Why did the right hemisphere decreased fNIRS signals? Are there any differences in function between hemispheres in balancing tasks?
Author Response

(The authors gave the same response as above.)

Round 2
Reviewer 3 Report
Thank you for valid comments.
Authors should correct citation errors in previous studies.
Reference number: 1,7,11,14,23,39,44,47,49